# SARS-CoV-2 Spike Protein 1 Causes Aggregation of α-Synuclein via Microglia-Induced Inflammation and Production of Mitochondrial ROS: Potential Therapeutic Applications of Metformin

**DOI:** 10.3390/biomedicines12061223

**Published:** 2024-05-31

**Authors:** Moon Han Chang, Jung Hyun Park, Hye Kyung Lee, Ji Young Choi, Young Ho Koh

**Affiliations:** Division of Brain Diseases Research, Department of Chronic Disease Convergence Research, Korea National Institute of Health, 187 Osongsaengmyeong2(i)-ro, Osong-eup, Heungdeok-gu, Cheongju-si 28159, Republic of Korea; def9207@gmail.com (M.H.C.); ntpeace@korea.kr (J.H.P.); lhk215@korea.kr (H.K.L.); jiyoung0220@gmail.com (J.Y.C.)

**Keywords:** α-synucleinopathy, long COVID, SARS-CoV-2 Spike protein 1, neuroinflammation, microglia, dopaminergic neuron, mitochondrial ROS, neurotoxin MPP+, metformin, neurodegenerative diseases

## Abstract

Abnormal aggregation of α-synuclein is the hallmark of neurodegenerative diseases, classified as α-synucleinopathies, primarily occurring sporadically. Their onset is associated with an interaction between genetic susceptibility and environmental factors such as neurotoxins, oxidative stress, inflammation, and viral infections. Recently, evidence has suggested an association between neurological complications in long COVID (sometimes referred to as ‘post-acute sequelae of COVID-19’) and α-synucleinopathies, but its underlying mechanisms are not completely understood. In this study, we first showed that SARS-CoV-2 Spike protein 1 (S1) induces α-synuclein aggregation associated with activation of microglial cells in the rodent model. In vitro, we demonstrated that S1 increases aggregation of α-synuclein in BE(2)M-17 dopaminergic neurons via BV-2 microglia-mediated inflammatory responses. We also identified that S1 directly affects aggregation of α-synuclein in dopaminergic neurons through increasing mitochondrial ROS, though only under conditions of sufficient α-Syn accumulation. In addition, we observed a synergistic effect between S1 and the neurotoxin MPP+ S1 treatment. Combined with a low dose of MPP+, it boosted α-synuclein aggregation and mitochondrial ROS production compared to S1 or the MPP+ treatment group. Furthermore, we evaluated the therapeutic effects of metformin. The treatment of metformin suppressed the S1-induced inflammatory response and α-synucleinopathy. Our findings demonstrate that S1 promotes α-synucleinopathy via both microglia-mediated inflammation and mitochondrial ROS, and they provide pathological insights, as well as a foundation for the clinical management of α-synucleinopathies and the onset of neurological symptoms after the COVID-19 outbreak.

## 1. Introduction

Synucleinopathies (also known as α-synucleinopathies), which encompass typical forms of Parkinson’s disease (PD), Lewy body dementia, and multiple system atrophy (MSA), are neurodegenerative diseases characterized by the abnormal accumulation of α-synuclein (α-Syn) protein aggregates within nerve cells or nerve fibers [1,2]. Reports of α-synucleinopathy in both familial and sporadic cases, independent of mutated genes, suggest that α-Syn aggregation is a common feature of both familial and sporadic diseases and is involved in disease onset and progression [3,4]. Most α-synucleinopathy cases are sporadic, and their onset has been linked to an interaction between genetic susceptibility and environmental factors such as neurotoxins, inflammation, and viral infections [5]. In particular, pandemic-scale outbreaks of infectious diseases have been implicated in the onset of α-synucleinopathies, represented by PD, given the scale and scope of infection [6,7]. Clinical case reports have revealed an increase in parkinsonism diagnoses following the Spanish flu pandemic of 1918 [8,9]. In addition, many in vitro and in vivo studies have suggested that viruses, including influenza, increase the aggregation rate of α-Syn [10,11].

The novel coronavirus disease 2019 (COVID-19) pandemic, caused by severe acute respiratory syndrome coronavirus 2 (SARS-CoV-2), has claimed millions of lives worldwide [12]. Although infection rates have decreased since the beginning of the outbreak, emerging evidence points to the presence of long-term, multi-organ clinical sequelae, referred to as long COVID, which presents a new challenge. Long COVID is diagnosed in patients that do not recover within 7 months after infection; most continue to experience systemic and neurological symptoms, even after the virus becomes undetectable in their system [13]. Notably, long COVID is commonly manifested as brain-related sequelae such as brain fog, anxiety, sleep disorders, cognitive problems, and movement disorders, which overlap with motor and non-motor syndromes in patients with α-synucleinopathies [14,15,16]. In addition, the worsening of motor and non-motor symptoms in patients with PD was reported during the acute phase of the SARS-CoV-2 infection [15,17,18,19]. Overall, a growing body of evidence seems to link long COVID and α-synucleinopathy; however, its precise pathology is largely unknown.

Significant increases in neuroinflammation in a wide range of brain regions were reported in patients with long COVID compared to controls. These regions included the middle and anterior cingulate cortex, corpus callosum, thalamus, basal ganglia, and ventricular border [20]. However, several hypotheses aimed to elucidate the development of brain-related symptoms and inflammation in patients with long COVID despite the absence of the virus. Recent reports proposed that spike protein 1 (S1) of SARS-CoV-2 persisted in the blood and brains of patients for 2 to 15 months or even longer after the initial infection, highlighting it as a major contributor to the neuroinflammation and brain-related sequalae of long COVID [21,22,23]. Fontes-Dantas et al. reported that injecting S1 into the brains of mice induces gliosis and late cognitive dysfunction similar to the neurological sequelae of long COVID and that the TLR4-2604G>A GG genotype is associated with poorer cognitive function in patients with long COVID [24]. It has also been reported that S1 increases the hippocampal neuronal cell death rate by activating primary microglial cells and inducing cognitive deficits and anxiety-like behaviors in rodent models [25]. Also, neuroinflammation appears to play an important role in the development and progression of α-synucleinopathy [26]. Considering inflammation as a key driver of α-Syn abnormal aggregation [27], this evidence strongly suggests that S1-induced neuroinflammation may contribute to α-synucleinopathy. The imbalance between the mutual regulation of mitochondria and α-synuclein leads to neuronal damage and contributes to the development of α-synucleinopathy [28]. S1 has been reported to cause mitochondrial dysfunction, altering Δψm, mCa2+ overload, reactive oxygen species (ROS) accumulation, and mitochondrial dynamics in human cell lines and rodent models [29,30,31]. Reports also indicate that S1 aggregates α-Syn directly via induction of mitochondrial ROS in neuronal cell lines [32,33]. Overall, evidence suggests that S1 can cause α-synucleinopathy, which may be the pathological mechanism underlying neurological symptoms or the onset of α-synucleinopathies after COVID-19 outbreak.

Metformin, dimethyl biguanide, is chemically synthesized from guanidine. It has its origin in an alkaloid called isoamylene guanidine, which was isolated from French lilac (Galega officinalis) [34,35]. Metformin is a common treatment for type 2 diabetes; it has been shown to possess protective properties via regulating oxidative stress and inflammatory response [36]. Recent studies indicated that metformin can transport across the blood–brain barrier and confer neuroprotection against various neurodegenerative diseases, including PD [37,38,39,40]. A recent meta-analysis showed that a combination treatment of metformin and sulfonylurea significantly reduces risk of PD in type 2 diabetes patients [41]. Furthermore, animal studies demonstrated that metformin protects the neurons from 1-methyl-4-phenyl-1,2,3,6-tetrahydropyridine (MPTP)-induced neurotoxicity [42]. In this study, therefore, we assessed the neuroprotective effects of metformin against S1-induced α-synucleinopathy.

This study aimed to investigate if S1 induces α-synucleinopathy and examine the mechanisms involved both in vivo and in vitro. It is the first to show that intranasal administration of S1 increases the aggregation of α-Syn in the brain of a rodent model. In vitro, we showed that S1 induced a pro-inflammatory response in BV-2 microglial cells and mitochondrial ROS production in BE(2)M-17 dopaminergic neurons, leading to the aggregation of α-Syn. Furthermore, building on previous findings, we showed the synergistic effects of MPP+, an environmental neurotoxin implicated in PD; the aggregation of α-Syn; and the therapeutic effects of metformin used as an anti-inflammatory and anti-mitochondrial damage agent.

## 2. Material and Method

### 2.1. Reagents

JSH-23 (ab144824) was purchased from Abcam (Minneapolis, MN, USA). JC-1 Dye (T3168) and MitoSox Red dye (M36008) were obtained from Invitrogen (Carlsbad, CA, USA. MPP+ iodide (D048), metformin (317240), and MitoTEMPO (SML0737) were purchased from Sigma (Saint Louis, MO, USA). The SARS-CoV-2 spike S1 subunit protein (10522-CV) was purchased from R&D Systems (Minneapolis, MN, USA). Human WT α-synuclein cloned into the pHM6 (40824) was obtained from Addgene (Cambridge, MA, USA). The antibody information is listed in Appendix A.

### 2.2. Animals and SARS-CoV-2 Spike Protein 1 Administration

Male Sprague–Dawley rats (6 weeks old, 180–210 g) were obtained from Samtako (Samtako, Gyeonggi, Republic of Korea). Rats were housed in a room at 21–25 °C in a 12 h light–dark cycle. All rats had access to food and water ad libitum. The animal protocol used was reviewed and approved by the KCDC-Institutional Animal Care and Use Committee (KCDC-IACUC; Approval Number KCDC-IACUC-22-008). After 2 weeks of adaptative feeding, rats were randomly divided into two groups: a S1-injected group and a sham control group (0.01 M PBS injected). The rats were anesthetized with an intramuscular injection of ketamine/xylazine hydrochloride (2:1) (100–150 μL/100 g body weight). The body temperature was maintained at 37 °C using a heating pad. The nasal administration procedure was carried out in accordance with the methodology previously described by Thorne et al. (2004) [32]. In brief, a nasal drop containing S1 protein suspension (0.5 μg/10 μL) or PBS (10 μL) was carefully placed on one nostril of an anesthetized animal (supine 90° angle) using a sterile 26-G Hamilton microsyringe (80330; Hamilton Company, Reno, NV, USA) and allowed to be delivered to the nasal cavity.

### 2.3. Immunohistochemistry

Brains were fixed in 4% paraformaldehyde solution, paraffin-embedded, and sectioned at a thickness of 8 μm. Brain sections were dried for 1 h at 65 °C, deparaffinized, rehydrated, and subjected to target retrieval using the retrieval solution (Dako, Glostrup, Denmark) manufacturer’s instructions. Sections were then blocked with blocking solution for 1 h at room temperature. Primary antibodies anti-Iba-1 (FUJIFILM Wako Pure Chemical Corporation, Osaka, Japan) and anti-α-synuclein (5G4) (Sigma-Aldrich, St. Louis, MO, USA) were used at a concentration of 1:100. After incubation with primary antibodies, brain sections were washed with PBS and incubated with Cy5 labeled anti-rabbit IgG (1:200; Jackson ImmunoRes, West Grove, PA, USA) or FITC labeled anti-mouse IgG (1:200; Jackson ImmunoRes, West Grove, PA, USA) secondary antibody. The sections were then mounted with mounting solution containing DAPI (Vector Laboratories, Peterborough, UK). Fluorescent images were obtained by using the EVOS M5000 imaging system (FITC: 510 nm emission wavelength, Cy5: 593 emission wavelength). For detecting α-synuclein or tyrosine hydroxylase (TH) in substantia nigra, primary antibody anti-α-synuclein (5G4) (1:100, Sigma-Aldrich, St. Louis, MO, USA) or anti-TH (1:200, Abcam, Cambridge, UK) was used and then incubated with biotinylated anti-mouse IgG (1:200; Dako, Glostrup, Denmark). The sections were rinsed, and an avidin and biotinylated horseradish peroxidase complex was applied for 1 h at room temperature (Vectastain Elite ABC reagent kit; Vector Laboratories, Newark, CA, USA). Diaminobenzidine (ImmPACT DAB substrate; Vector Laboratories, Newark, CA, USA) was applied for 2 min to detect the target protein. The sections were imaged using a microscope (Olympus, Tokyo, Japan).

### 2.4. BE(2)-M17 Cell Culture

BE(2)-M17 cells (CRL-2267) were purchased from ATCC (Manassas, VA, USA). Cells were grown in Dulbecco’s modified Eagle’s medium and Nutrient mixture F12 (DMEM/F12) supplemented with 10% fetal bovine serum and 1% penicillin/streptomycin. Cells were maintained at 37 °C in humidified atmosphere with 95% air and 5% CO2. Cells were pretreated with the MitoTEMPO and Metformin for indicated time before the treatment with SARS-CoV-2 spike S1 and MPP+ at the designated concentrations and hours. Negative control was used as control group.

### 2.5. BV-2 Microglial Cell Culture and Conditioned Media (CM) Preparation

BV-2 (ATCC) cells were cultured in DMEM supplemented with 5% FBS and 1% P/S in a 60 mm cell culture dish at 37 °C in a humidified atmosphere of 95% air/5% CO_2_. For the experiments, cells were seeded in a 60 mm cell culture dish at 0.3 × 106 cells per dish or in 96-well cell culture plates at 0.5 × 104 cells per well. They were cultured overnight. Cells were then incubated for 24 h in serum-free DMEM (untreated controls) or serum-free DMEM containing S1 (200 ng/mL), with or without 5 μM JSH-23 and 0.5 mM metformin. The next day, the medium was removed, and cultured cells were washed with PBS once to remove the S1, JSH-23, and metformin. Then, the cells were further cultured for 24 h in the same culture medium to collect CM. The medium was collected and centrifuged at 3000 × rcf for 10 min to remove cell debris.

### 2.6. Immunoblot

The rat brain and cells grown in plates were washed with phosphate-buffered saline (PBS) and lysed in RIPA buffer. The cell lysate was then centrifuged (13,000 rpm, 30 min, 4 °C), and the protein concentration was determined using the Bradford protein assay, in accordance with the manufacturer’s instructions. The proteins were separated using a 4–12% NuPAGE gel (Invitrogen) and transferred onto nitrocellulose membranes. The membranes were then blocked in tris-buffered saline with 5% skim milk and 0.1% Tween 20 and incubated with the primary antibody overnight at 4 °C. Subsequently, the membranes were incubated with a secondary antibody, following a series of washes. The proteins were then detected using a chemiluminescent kit (Amersham Pharmacia Biotech, Buckinghamshire, UK, USA). Detection of aggregated α-synuclein was followed as described [33]. Membranes were placed into 0.4% PFA fixative solution for 30 min at room temperature. Membranes were slowly shaken on a rotator to ensure full coverage.

### 2.7. Multiplex Analysis of Cytokines in BV-2 CM

Profiling of pro-inflammatory cytokines in BV-2 CM was performed using multiplexing xMAP technology (Luminex Corp., Austin, TX, USA) according to the manufacturer’s instructions. Mouse cytokines were analyzed using a 17-plex kit for TNFSF23B, CCL5, Chitinase 3-like 1, IFN-γ, IL-1b, IL-2, IL-4, IL-6, IL-10, IL-27, IL-6R, TNF-α, TNFRI, TNFRII, and TWEAK (R&D Systems, Minneapolis, MN, USA). The concentration of each cytokine was quantified using standard curves for the respective standard materials.

### 2.8. Cell Transfection

Cells were transiently transfected with the human WT α-synuclein cloned into the pHM6-his-tag vector. The Lipofectamine LTX reagent and Opti-MEM medium (Life Technology, Grand Island, NY, USA) were used for transfection according to the manufacturer’s instructions.

### 2.9. Measurement of Mitochondrial Redox Status and Damage

Healthy mitochondrial membrane potentials were detected using the fluorescent probe JC-1 (Invitrogen). Cells were incubated at 37 °C with 5 μM JC-1 for 30 min. The ratio of the intensity of green/red fluorescence was directly proportional to the mitochondrial membrane potential. The mitochondrial ROS level was measured using MitoSox. Cells were treated for 30 min with 5 μM of MitoSox. Green (excitation/emission: 500/536 nm) and red fluorescence (excitation/emission: 510/580 nm) were imaged on a fluorescence microscope (Zeiss, Jena, Germany).

### 2.10. Statical Analysis

The statistical analysis of the data was conducted using GraphPad Prism v7.0. The values are expressed as the mean ± standard error of the mean (SEM). Student’s *t*-test was employed to ascertain the statistical significance of the differences between the two samples. A *p*-value of less than 0.05 was used as the criterion for statistical significance.

## 3. Results

### 3.1. SARS-CoV-2 S1 Increases α-Syn Aggregation and Microglial Cell Activation in Rat Brain

To assess the effect of S1 on α-synucleinopathy at the organ level, 0.5 g single doses of S1 or PBS were administered intranasally to 8-week-old male rats, and the brains were collected 6 weeks after administration. (Figure 1A). Intranasally administered S1 has been reported to enter all brain regions (olfactory bulb, striatum, FCtx, PCtx, OCtx, hypothalamus, hippocampus, thalamus, cerebellum, midbrain, and pons/medulla) [36]. To validate this, His-tagged S1 was used. His-tagged S1 was detected via Western blot analysis using a His protein-specific antibody in the striatum of rat brain (Appendix A). 

The abnormal accumulation of α-Syn aggregates in dopaminergic neurons is the most characteristic feature of synucleinopathy. To assess the aggregation of α-Syn, histological analysis with DAB staining was performed using an aggregated α-Syn (5G4)-specific antibody in substantia nigra (SNpc). We observed higher intracellular aggregated α-Syn (5G4) signals in the SNpc region of the S1 group than in the sham group (Figure 1B). In addition, we performed Western blot analysis using aggregated α-Syn (5G4) antibody and observed significantly increased aggregated form levels of α-Syn in the S1-injected striatum of rat brain (Figure 1C,D). Neuroinflammation, a key factor in α-synucleinopathies exacerbated by SARS-CoV-2, is predominantly regulated by microglia in the brain [20,43,44,45]. Immunofluorescence analysis showed that the Iba-1 signal, which is a marker of activated microglia, was more abundant in the SNpc of the S1 group than in the sham group (Figure 1E). Interestingly, activated microglia (Iba-1) were more frequently observed in regions enriched with aggregated 5G4 α-Syn in the SNpc (Figure 1E). Collectively, our results indicate that intracerebral administration of S1 induces typical pathophysiological features of synucleinopathy, such as α-Syn aggregation and brain inflammation [45], which led us to further assess mechanisms of S1-induced α-Syn aggregation in vitro.

### 3.2. SARS-CoV-2 S1 Increases Aggregation, Phosphorylation, and Monomer Levels of α-Syn via the Microglial Pro-Inflammatory Response

Pro-inflammatory cytokines secreted by microglia are strongly implicated in the mechanisms of α-Syn aggregation [27,44]. Furthermore, we observe in Figure 1E that histological analysis of the SNpc region of the brain from S1-treated rats showed activated microglia surrounding the syn aggregate signal. Therefore, to identify whether S1 affected aggregation of α-Syn via a pro-inflammatory response induced by microglia activation, conditioned media (CM) from BV-2 microglial cells were used to treat the BE(2)M-17 dopaminergic neuronal cell line. To verify whether the change in BE(2)M-17 via CM of BV-2 was due to an inflammatory response, we used JSH-23, an NF-kB inhibitor, which is a typical inflammatory regulator. The study design is shown in Figure 2A. 

First, we observed the effects of S1 on the aggregation of α-Syn. Aggregated α-Syn (5G4) was significantly increased by S1-treated BV-2 CM, which was restored by JSH-23 (Figure 2B,C). Phosphorylation of α-Syn is an important pathological factor in abnormal aggregation of α-Syn; approximately 90% of α-Syn deposited in Lewy bodies is phosphorylated at Ser129 in patients with PD [38]. The same trend was observed for *p*-α-Syn (Ser129) and monomeric α-Syn (Figure 2B,D,E). We then examined the pro-inflammatory cytokine profiles in BV-2 CMs to determine whether the cytokines were involved. CCL5, IL-10, and TWEAK levels were significantly increased by S1 treatment, whereas only CCL5 and TWEAK levels were significantly restored by JSH-23 treatment (Figure 2F). These profiling data suggest that the increased aggregation of α-Syn via S1-treated BV-2 CM in BE(2)M-17 is likely affected by CCL5 and TWEAK. In general, microglia respond to targets by becoming activated and polarized into a typical M1-like (pro-inflammatory) or M2-like (anti-inflammatory) phenotype [43]. Pro-inflammatory cytokines and Iba-1 are classic markers of the M1-like phenotype, whereas CD163 is a marker of the M2-like phenotype [43]. Western blot analysis of BV-2-cell lysates confirmed that S1 significantly increased the protein levels of TNF-α, IL-1β, and Iba-1, while CD163 remained unchanged (Figure 2G–K). These results indicate that S1 converts BV-2 microglial cells into a pro-inflammatory M1-like phenotype. Overall, our results clearly indicate that the inflammatory response of microglial cells triggered by S1 increases the aggregation of α-Syn in dopaminergic neurons via both inducing phosphorylation and increasing protein levels of α-Syn.

### 3.3. SARS-CoV-2 S1 Directly Increases Aggregation of α-Syn Only if α-Syn Levels Are Sufficient

To determine whether S1 directly affects dopaminergic neurons, we treated BE(2)M-17 cells with S1. The aggregated form of α-Syn (5G4) and monomeric α-Syn was not altered by S1; however, p-α-Syn (Ser129) was significantly increased by S1 (Appendix A). It is well established that Ser129 phosphorylation of α-Syn increases α-Syn aggregation [46]. Thus, this result suggests that when S1 acts independently on neurons, it has the potential to increase α-Syn aggregation, but this effect may occur only if sufficient α-Syn is accumulated. Therefore, we aimed to observe the direct effects of S1 in BE(2)M-17 overexpressing α-Syn. Several studies have reported that S1 increases mitochondrial ROS levels and damage [29,47]. Also, mitochondrial damage is associated with the aggregation of α-Syn [48]. We therefore used MitoTempo, a mitochondria-specific ROS scavenger, to verify whether S1 directly promotes aggregation of α-Syn in BE(2)M-17 via mitochondrial ROS production.

As shown in Figure 3A,B, we observed that the level of aggregated α-Syn (5G4) was increased by S1 and restored by MitoTempo. To verify whether this effect was linked to mitochondrial damage, we observed fluorescence images using JC-1, a probe specific to mitochondrial damage. The results show that the whole mitochondria (green) signal was unchanged by S1 treatment, whereas the damaged mitochondria (red) signal was increased, which was reversed by MitoTempo treatment (Figure 3C,D). Finally, to verify S1-induced mitochondrial ROS production, we observed fluorescence images using MitoSox, a mitochondrial ROS-specific dye. Mitochondrial ROS (red) levels were increased by S1 and alleviated by MitoTempo (Figure 3E,F). We concluded that only under conditions of sufficient α-Syn accumulation does S1 directly aggregate α-Syn by increasing mitochondrial damage and ROS production in neuron cells.

### 3.4. SARS-CoV-2 S1-Induced Microglial Activation Enhances MPP+-Induced Cytotoxicity and α-Syn Aggregation and the Attenuating Effects of Metformin

Most cases of α-synucleinopathy are sporadic, and it is well known that there is an interaction between different factors, such as neurotoxins, inflammatory factors, and genetic susceptibility [4]. Therefore, we used the neurotoxin MMP+, a typical environmental factor that causes α-synucleinopathies [49], to examine whether it has a synergistic effect with other environmental factors and S1. CM from BV-2 microglial cells and MPP+ were used to treat the BE(2)M-17 dopaminergic neuronal cell line. The study design is shown in Figure 4A. First, the MTT assay was performed to observe the synergistic effects of S1 and MPP+ on the viability of dopaminergic neurons. As shown in Figure 4B, S1-treated BV-2 CM with MPP+ decreased cell viability to a greater extent than either treatment alone. No change in viability was observed when 250 μM MPP+ was used alone (white bar); thus, we continued to observe the synergistic effect on the aggregation of α-Syn under this condition (Figure 4B). We observed that the aggregated forms α-Syn (5G4) and p-α-Syn (Ser129) were significantly increased in S1-treated BV2 CM and MMP+-treated BE(2)M-17 compared to their respective treatments, whereas no synergistic effect of the α-Syn monomer was identified (Figure 4C–F). These results indicate that the S1-induced microglial inflammatory response and MMP+ share a pathological mechanism for Ser129 phosphorylation of α-Syn in dopaminergic neurons. 

A recent clinical trial on long COVID reported significant symptom relief with metformin [50]. Metformin has also been reported to attenuate the inflammatory responses in the microglia in rodent models [51,52]. Therefore, to determine whether the modulation of S1-induced BV-2 activation by metformin affects dopaminergic neurons, we treated CM with metformin, S1, and MPP+. The study design is shown in Figure 4A. Metformin restored the decrease in cell viability and increase in the aggregated form of α-Syn (5G4), as well as p-α-Syn (Ser129) induced by S1-treated BV-2 CM with MPP+, whereas no change in syn-monomer level was observed (Figure 4B–F). We tested whether metformin attenuated S1-induced BV-2 activation. Pro-inflammatory cytokine profiling of BV-2 CM showed that metformin only attenuated CCL5 secretion (Figure 4G). As shown in Figure 4H–L, Western blot analysis of BV-2 cell lysates confirmed that metformin ameliorated the S1-induced M1-like phenotype (TNF-α, IL-1β, and Iba-1), while CD163 levels remained unchanged.

### 3.5. SARS-CoV-2 S1 Directly Enhances MPP+-Induced Cytotoxicity and α-Syn Aggregation, with Attenuating Effects of Metformin

To determine whether S1 has a direct synergistic effect with MPP+ in dopaminergic neurons, we used BE(2)M-17 cells overexpressing α-Syn. The MTT assay confirmed that 500 ng/mL S1 with 500 μM MPP+ significantly reduced cell viability, compared to either treatment alone (Figure 5A), and that metformin ameliorated these effects (Figure 5B); thus, we continued to observe the moderating effect of metformin on the S1-induced aggregation of α-Syn under this condition. The aggregated form of α-Syn (5G4) was increased by S1 with MPP+ compared to that with MPP+ alone (Figure 5C,D). Mitochondrial damage was detected using JC-1 dye, and the number of damaged mitochondria (red) was increased by S1 and MPP+ treatment compared to the control group (Figure 5E,F). In addition, image analysis of mitochondrial ROS using MitoSOX confirmed that the signal was increased by S1 and MPP+ treatment compared to the control (Figure 5G,H). Finally, we observed that metformin reversed all changes induced by S1 with MPP+ treatment. In summary, combined with Figure 4, we clearly demonstrated that S1 and the neurotoxin MPP+ have a synergistic effect on cytotoxicity and aggregation of α-Syn, and that metformin attenuates these effects.

## 4. Discussion

While the role of viral infections in the development of α-synucleinopathies remains controversial, growing evidence suggests a link between viral infections and disease onset [6,7,53]. Post-pandemic, many patients have presented with long COVID, including neurological complications, some of which may resemble Parkinson’s disease and other α-synucleinopathies, such as cognitive dysfunction, anxiety, brain fog, sleep disturbances, and movement disorder, suggesting a link between α-synucleinopathy and the neurological complications seen in long COVID [14,15]. However, evidence remains limited on the long-term clinical course of COVID-19, and any mechanisms involved are yet to be elucidated. Herein, we aimed to evaluate the association between long COVID and α-synucleinopathy and to describe the underlying mechanism in vivo and in vitro.

Our hypothesis is supported by recent findings that S1 persists in the plasma of patients with COVID-19 for up to 15 months after diagnosis and accumulates in the brain [21,22,23]. In particular, the recent findings by Fontes-Dantas et al. suggest a pathology for long COVID neurological complications: S1 affects the mouse central nervous system via mechanisms independent of the index viral infection, leading to long-term cognitive dysfunction 30–60 days but not 7 days after the injection of S1 [23]. In line with these observations, we found that aggregation of α-Syn in rat brain regions, including the striatum and SNpc, that was observed 6 weeks after an intranasal injection of S1. Growing evidence suggests that neuroinflammation is involved in the development of a variety of neurodegenerative diseases, including α-synucleinopathies such as PD, and is a major contributor to disease progression [4]. Neuroinflammation is also the most prominent feature of neuropathological mechanisms induced by SARS-CoV-2 infection [54,55]. A recent clinical study found persistent inflammation in the brain and blood samples of patients presenting with neurological symptoms of long COVID at 3–10 months post-infection, suggesting that neuroinflammation is an important factor in the neuropathological mechanism of long COVID [19]. Likewise, we identified S1-induced neuroinflammation in this study, associated with an increase in activation of microglia in the SNpc of the rat brain. Furthermore, we observed that activated microglial cells appeared in the vicinity of neurons with increased aggregation of α-Syn in the rat SNpc. Although our study did not consider the route or site of SARS-CoV-2 infection, to our knowledge, this is the first study to show that the presence of residual S1 in the brain can cause α-synucleinopathies in rodent models.

Microglial cells are key regulators of the immune system and inflammation in the brain [56]. Recent studies have demonstrated that microglial activation-induced pro-inflammatory cytokines are major contributors to the cognitive deficit induced by S1 in mouse models [23,24]. In this study, we showed that S1 turns BV-3 microglial cells into M1 phase-like phenotypes and that its CM increases the levels of the aggregated form of α-Syn in the BE(2)M-17 dopaminergic cell line. In addition, we confirmed that the observed phenomenon was due to an inflammatory response using JSH-23, an inhibitor of NF-kB, one of the most prominent targets of inflammatory regulation. We then profiled pro-inflammatory cytokines in the CM of BV-2 cells and found that CCL5 and the TNF family member TWEAK levels were significantly elevated by S1 and rescued by JSH-23. Clinical studies have reported that CCL5 and TNF-α persist in the blood of patients with long COVID, as well as PD [57,58,59,60]. In a mouse model, CCL5 was reported to increase the deposition of α-Syn in the substantia nigra and aggregate prion-like protein [61,62]. TNF-α secreted by mouse primary microglial cells promotes the propagation of α-Syn in neuronal cells [63]. In line with these observations, we found that CM of BV-2 cells increases the protein levels of the α-Syn monomer. Therefore, our findings suggest the importance of the CCL5 and TNF family, such as TWEAK, in long COVID. Collectively, we demonstrated that microglial cells activated by S1 promote α-Syn aggregation via the secretion of pro-inflammatory cytokines and provide novel pathological mechanisms and insight into neurological complications and α-synucleinopathy risk associated with long COVID.

Recently, Wang et al. and Wu et al. reported that S1 aggregates α-Syn via mitochondrial damage and Ser129 phosphorylation [30,31]. However, both studies have limitations in proving this hypothesis, as they used a single neuronal cell line and the experiments involved artificial overexpression of both S1 and α-Syn, which fails to reflect the systemic environment and transmembrane dynamics of circulating S1 in an actual brain. Herein, we identified the aggregation of α-Syn in BE(2)M-17 dopaminergic cell lines via direct treatment of S1 rather than via intracellular overexpression. Also, we did not observe any aggregation of α-Syn from S1 treatment when α-Syn was not overexpressed. This suggests the direct neuronal effects of S1 require sufficient intracellular α-Syn to be activated. However, α-Syn aggregation via microglial activation was observed without α-Syn overexpression in neurons. Overall, we believe that our study allows for a broader understanding and validation of this hypothesis, and we cautiously predict that in the actual brain, microglia cell activation and inflammation from S1 likely contribute more significantly to α-synucleinopathies than the direct effects on neurons.

Most α-synucleinopathy cases occur sporadically, and it is generally known that an interaction between genetic susceptibilities and environmental factors such as neurotoxins, inflammation, and infection with viruses contributes to the development of the disease [4]. In this context, neurological sequelae of long COVID do not present in all infected individuals, indicating the potential for an environmental or genetic factor to which the patient may have been exposed, sharing pathological pathways with S1 [64]. Additionally, in our experiment, Th-positive dopaminergic cell counts in the substantia nigra (Appendix A) were not affected by the administration of S1. MPTP is a precursor of the neurotoxin MPP+, which causes permanent symptoms of PD by destroying dopaminergic neurons in the substantia nigra in the brain [65]. MPP+ has been reported to promote the expression and aggregation of α-Syn and has a synergistic effect with the influenza virus [49,66], suggesting a pathological overlap between S1 and MPP+. Herein, we demonstrated the synergistic effect of S1 on MPP+-induced cell death and α-Syn aggregation in dopaminergic cell lines via BV-2 microglial cell-induced inflammatory response and mitochondrial damage. This finding provides insight into the etiology of α-synucleinopathy, such as PD, onset after COVID-19 outbreak, depending on the presence of SARS-CoV-2 infection.

Metformin is used worldwide to treat patients with type 2 diabetes mellitus and polycystic ovary syndrome. Recently, in vitro, ex vivo, and randomized clinical studies have reported that metformin reduces SARS-CoV-2 infection [67,68,69,70]. Also, outpatient treatment with metformin has been shown to reduce long COVID incidence by approximately 41% compared with placebo treatment in a multicenter, randomized, quadruple-blind, parallel-group phase 3 trial [50]. Metformin also reduces microglial activation and repairs mitochondrial damage in neuronal cells [51,52,71]. Thus, these studies strongly support the inhibitory effect of metformin on α-Syn aggregation by S1, which is consistent with this study. Herein, we showed that metformin attenuates cell death and α-Syn aggregation induced by S1 and MPP+. Metformin-treated CM of BV-2 microglial cells downregulated Ser129 phosphorylation of α-Syn in the dopaminergic cell line, but not the protein level of the α-Syn monomer. In addition, our pro-inflammatory cytokine profiling data on BV-2 CM revealed that metformin significantly reduces S1-induced CCL5 but not TWEAK levels. These results are consistent with reported findings, showing that CCL5 is involved in aggregation and the TNF family is involved in the intracellular protein levels of α-Syn [61,62,63]. Moreover, we observed that metformin directly attenuates S1-induced cell death and α-Syn aggregation in dopaminergic cell lines. Collectively, our findings suggest both direct and indirect modulatory effects of metformin on S1-induced α-synucleinopathies and may contribute to the clinical management of long COVID with neurological sequelae and α-synucleinopathies, including PD onset following the COVID-19 outbreak.

In conclusion, our findings reveal that S1 increases microglia-mediated inflammation and mitochondrial damage, leading to the aggregation of α-Syn by both inducing protein levels and Ser129 phosphorylation of α-Syn. These results highlight the role of S1 in the pathological mechanisms underlying α-synucleinopathies and the neurological symptoms of patients with long COVID. Moreover, we report the synergism between MPP+ and S1 and the attenuating effect of metformin, which may have implications for the treatment of α-synucleinopathies, including those linked to COVID-19 (Figure 6).

## Figures and Tables

**Figure 1 biomedicines-12-01223-f001:**
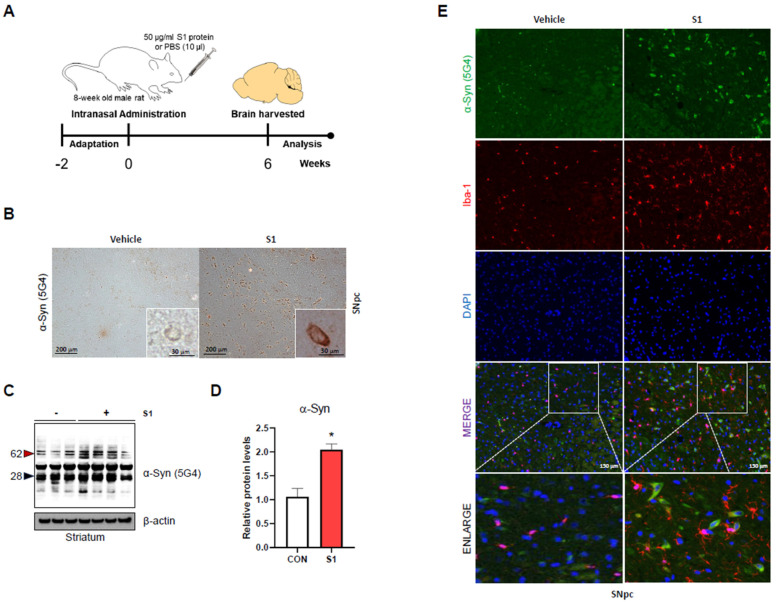
SARS-CoV-2 S1 increases α-Syn aggregation in a rat brain. (**A**) Experimental timeline for SARS-CoV-2 treatment. A single dose (0.5 g of S1 or PBS) was intranasally administered to 8-week-old male rats, and brains were collected at 6 weeks after administration. (**B**) Representative images of immunohistochemistry using aggregated α-Syn (5G4) specific antibody in substantia nigra. (**C**) Representative immunoblot images for protein levels of aggregated α-Syn (5G4). (**D**) Western blot quantification for protein levels of aggregated α-Syn (5G4) in rat brain striatum. (**E**) Immunofluorescence staining of Iba-1, representing markers for active microglia cells, and aggregated α-Syn (5G4) in substantia nigra of vehicle and S1-treated rat brain. (Green: α-Syn (5G4) and red: Iba-1) (Scale bar: 100 μm). Data are presented as mean ± standard error of the mean (*n* = 3–4 mice per group, * *p* < 0.05).

**Figure 2 biomedicines-12-01223-f002:**
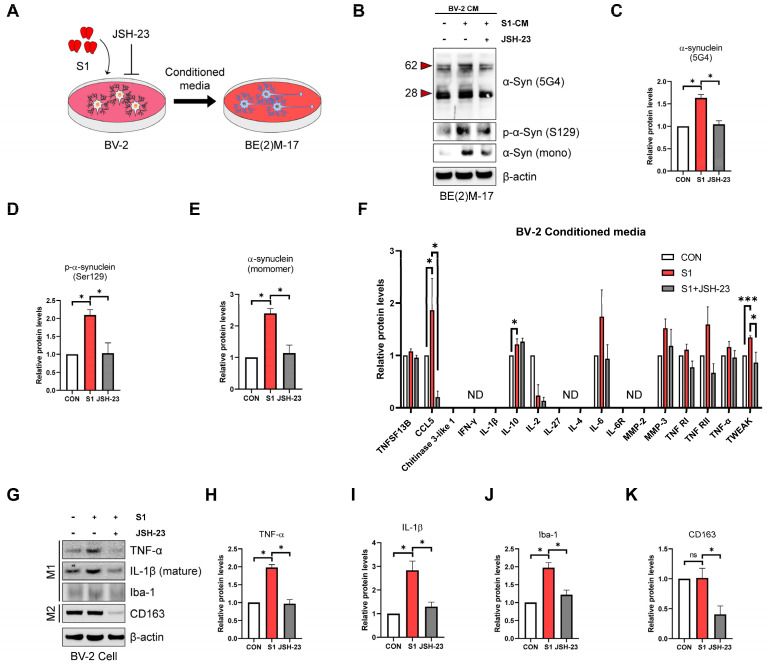
SARS-CoV-2 S1 increases aggregation and monomers of α-Syn via a microglia pro-inflammatory response. (**A**) A schematic diagram showing BE(2)M-17 dopaminergic neuron cells stimulated by S1 protein-activated glia-conditioned media (CM). BV-2 cells were incubated for 24 h in serum-free DMEM (untreated controls) or serum-free DMEM containing S1 (200 ng/mL) with or without 5 μM JSH-23. BE(2)M-17 cells were treated with collected BV-2 CM for 24 h. Further details are provided in Section 2. (**B**) Representative images of immunoblot analysis using specific antibodies for aggregated α-Syn (5G4), phospho-α-Syn (Ser129), and monomeric α-Syn. (**C**–**E**) Quantification of immunoblot analysis for aggregated α-Syn (5G4), phospho-α-Syn (Ser129), and monomeric α-Syn. (**F**) Multiplex cytokine analysis from conditioned media of BV-2 cells in response to S1 (200 ng/mL) with or without 5 μM JSH-23. (**G**) Representative images of immunoblot analysis using specific antibodies for TNF-α, mature IL-1β, Iba-1, and CD163. (**H**–**K**) Quantification of immunoblot analysis for TNF-α, mature IL-1β, Iba-1, and CD163. All data are presented as mean ± standard error of the mean (*n* = 3–5 per group, * *p* < 0.05 and *** *p* < 0.001, ND; Not Detected, ns; no significance).

**Figure 3 biomedicines-12-01223-f003:**
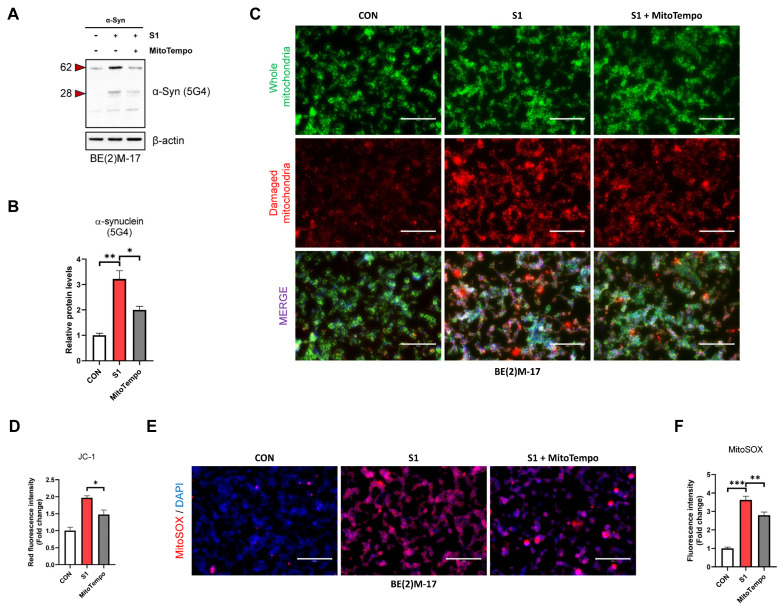
SARS-CoV-2 S1 directly increases aggregation of α-Syn above a threshold. α-Syn overexpressed BE(2)M-17 cells were incubated for 24 h in DMEM/F12 (untreated controls) or DMEM/F12 containing S1 (500 ng/mL), with or without 10 μM MitoTEMPO. (**A**) Representative images of immunoblot analysis using specific antibody for aggregated α-Syn (5G4) and (**B**) quantification of data. (**C**) Mitochondrial damage analyzed using JC-1 dye and (**D**) quantification data of JC1 determined using fluorescence microscope (green: whole mitochondria, red: damaged mitochondria). (**E**) Representative image of MitoSOX red staining for mitochondrial ROS and (**F**) quantification of data. All data are presented as mean ± standard error of the mean (*n* = 3–5 per group, * *p* < 0.05, ** *p* < 0.01, and *** *p* < 0.001). (Scale bar: 50 μm).

**Figure 4 biomedicines-12-01223-f004:**
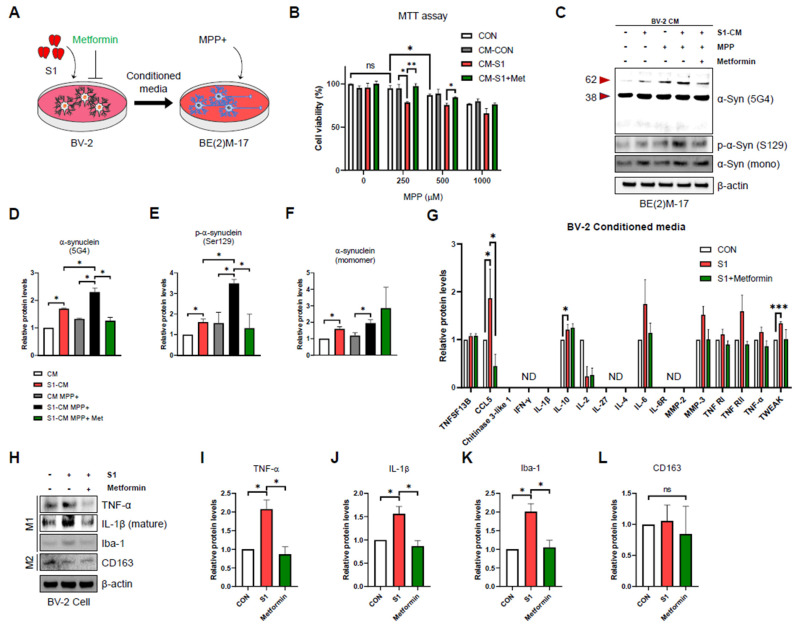
SARS-CoV-2 S1-induced microglial activation enhances MPP-induced cytotoxicity and α-Syn aggregation, with the moderating effects of metformin. (**A**) A schematic diagram showing BE(2)M-17 dopaminergic neuron cells stimulated by S1 protein-activated BV-2-conditioned media (CM). BV-2 cells were incubated for 24 h in serum-free DMEM (untreated controls) or serum-free DMEM containing S1 (200 ng/mL), with or without 1 mM metformin. BE(2)M-17 cells were treated with collected BV-2 CM with or without MPP+ for 24 h. Further details are provided in Section 2. (**B**) The effects of MPP+, 200 ng/mL S1-activated BV-2 CM, and metformin (alone and in combination) on the cell viability (% of control) of BE(2)M-17 cells for 24 h. (**C**) Representative images of immunoblot analysis using specific antibodies for aggregated α-Syn (5G4), phospho-α-Syn (Ser129), and monomeric α-Syn. (**D**–**F**) Quantification of immunoblot analysis for aggregated α-Syn (5G4), phospho-α-Syn (Ser129), and monomeric α-Syn. (**G**) Multiplex cytokine analysis from conditioned media of BV-2 cells in response to S1 (200 ng/mL) with or without 1 mM metformin. (**H**) Representative images of immunoblot analysis using specific antibodies for TNF-α, mature IL-1β, Iba-1, and CD163. (**I**–**L**) Quantification of immunoblot analysis for TNF-α, mature IL-1β, Iba-1, and CD163. All data are presented as mean ± standard error of the mean (*n* = 3–5 per group, * *p* < 0.05, ** *p* < 0.01, and *** *p* < 0.001, ND; Not Detected, ns; no significance).

**Figure 5 biomedicines-12-01223-f005:**
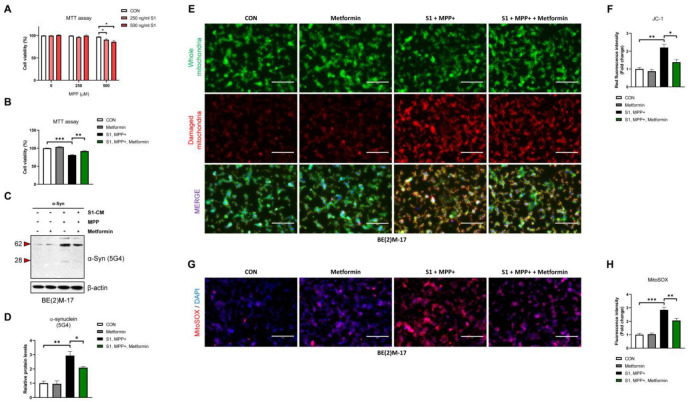
SARS-CoV-2 S1 directly enhances MPP-induced cytotoxicity, with moderating effects of metformin. (**A**) The effects of MPP+ (0, 250, 500 μM) and S1 (0, 250, 500 ng/mL) (alone and in combination) on the cell viability (% of control) of α-Syn-overexpressed BE(2)M-17 cells for 24 h. (**B**–**H**) α-Syn-overexpressed BE(2)M-17 cells were incubated for 24 h in DMEM/F12 (untreated controls) or DMEM/F12 containing S1 (500 ng/mL) with or without 1 mM metformin to identify moderating effects of metformin. (**B**) MTT assays were performed to identify moderating effects of metformin on S1 with MPP+ treatment-induced cytotoxicity. (**C**) Representative images of immunoblot analysis using specific antibody for aggregated α-Syn (5G4) and (**D**) quantification of data. (**E**) Mitochondrial damage analyzed using JC-1 dye and (**F**) quantification data of JC1 (green: whole mitochondria, red: damaged mitochondria). (**G**) Representative image of MitoSOX red staining for mitochondrial ROS and (**H**) quantification of data. All data are presented as mean ± standard error of the mean (*n* = 3–5 per group, * *p* < 0.05, ** *p* < 0.01, and *** *p* < 0.001) (Scale bar: 50 μm).

**Figure 6 biomedicines-12-01223-f006:**
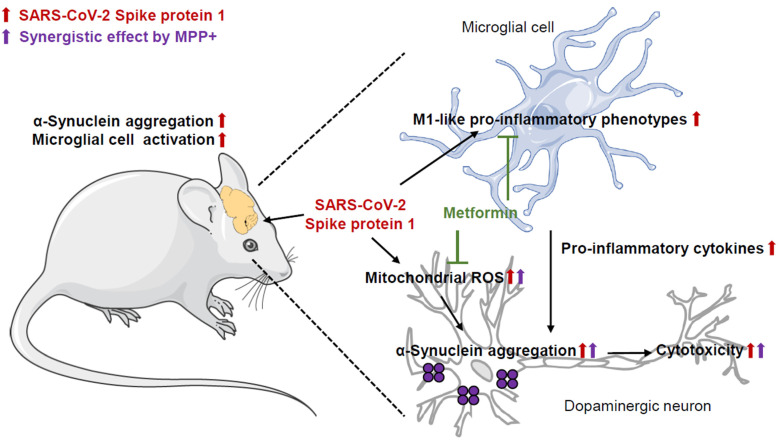
Proposal model of this study.

## Data Availability

The original contributions presented in the study are included in the article/Appendix A, further inquiries can be directed to the corresponding author.

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
