# Peer review of "SARS-CoV-2 Spike Protein 1 Causes Aggregation of α-Synuclein via Microglia-Induced Inflammation and Production of Mitochondrial ROS: Potential Therapeutic Applications of Metformin"

_biomedicines, 2024, doi:10.3390/biomedicines12061223_

Round 1

Reviewer 1 Report

Comments and Suggestions for Authors

Koh and coworkers reported the role and mechanism of SARS-CoV-2 spike protein 1 (S1) in long COVID-19 as well as the effect of metformin. The experimentation was well organized and their discussion was made carefully and logically. Thus, I believe that this manuscript could be published with minor changes raised below.

1. For fluorescent images, please show the excitation wavelengths.

2. Based on the mice test, is it possible to conclude the mechanism occurring in human infection? I suggest the authors to make a brief comment on it.

3. Have the authors consider the effect of third ingredient except for metformin?

Reviewer 2 Report

Comments and Suggestions for Authors

I have reviewed the paper by Chang et al.

The study is very thorough, and the data is solid.

I suggest removing in vivo and in vitro from the title which is already too long.

There is no mention of the effect of Metformin in the Abstract.

I would actually highlight it and would need to authors to contrast their findings between Figure 2 (H through K) using the NF-kB inhibitor JSH-23, and Figure 4 (I through L) using metformin. M2 microglia polarization marker CD163, does not change with Metformin but it decreases with JSH-23. Please explain this difference.

Is there a reason why authors do not use the term “polarization”?

Comments on the Quality of English Language

NA

Reviewer 3 Report

Comments and Suggestions for Authors Post-acute COVID syndrome (long-COVID) includes neurological complications, some of which may resemble Parkinson’s disease and other α-synucleinopathies as Lewy body dementia and multiple system atrophy, for which it is characteristic an accumulation of aggregated forms the intraneuron protein alpha-synuclein. On the other hand, in the literature it is described a persistence of S-protein of coronavirus in the blood and brain of patients with long-COVID for 2 to 15 months after the initial infection. The purpose of the research under review is to investigate if the water soluble S1-subunit of S-protein induces aggregation of the alpha-Syn in the neurons, as it appears in the cases of neurodegenerative diseases such as Parkinson's. Additionally, the authors try to disclose the mechanism of the S1-effect considering the literature data that S1 causes dysfunction of the mitochondria by altered transmembrane potential, overload of Ca2+ cations, and accumulation of reactive oxygen species (ROS).  The authors have carried out in-vivo experiments with rats and in-vitro with cell cultures of human and rodent dopaminergic neuronal cells, applying modern immunohistochemival techniques as immunoblot and Western blot quantification analyses using fluorescently labeled antibody which is specific to aggregated alpha-Syn. The analysis shows 2 times increased level of the aggregated intracellular alpfa-Syn in substantia nigra of rat brain after intracerebral injection with S1 protein. The authors infer that S1 increases the level of aggregated α-Syn in the dopaminergic neurons via the pro-inflammatory activation of the microglial cells, which leads to both increasing of the intracellular concentration of monomeric alpha-Syn and to its aggregation caused by phosphorylation of the Ser-129 aminoacid residue (according to the literature data). The microglial cells, activated by S1-protein, promote α-Syn aggregation via the secretion of pro-inflammatory cytokines. The aggregative effect of S1 coronavirus protein emerges only when α-Syn is overexpressed in the neuronal cells. The research is very well carried out on the experimental level, and the manuscript is well written. However, the authors are encouraged to clear up the next questions. 1. The authors assert that the intranasal administration of S1 increases the aggregation of alpha-Syn in the rat brain. This could be real if they infect the rats by intact coronavirus which can be reproduced in the epithelial cells and then reaches the glial and neuronal cells in the brain by blood circulation. However, S1 is a globular protein which can not be reproduced without RNA; S1 could cause only an immunological reaction which strongly decreases its concentration. How the limited quantity of S1 macromolecules introduced in the nasal epithelia causes aggregation of alpfa-Syn in the neuronal cells of the brain? 2.  The immunofluorescence intensity in the control and the 2 times higher one in the probe can be interpreted by two ways: (a) two times increased quantity of aggregated alpfa-Syn, or (b) two times increased quantity of single alpfa-Syn macromolecules. The authors’ interpretation is based on the assumption that the antibodies are specifically bounded only to the aggregated alpfa-Syn, but authors do not give any data about the affinity of the used antibody to the single and to the aggregated alpfa-Syn. It seems that the Fig.1D presents the fluorescence intensity but on the ordinate it is written “protein level”. This could be correct if (a) the constants (absolute or relative) of antibody binding to single and aggregated alpfa-Syn protein and (b) the molar ratio of antibody to alpfa-Syn in the aggregate are known. Then, the share of aggregated alpfa-Syn can be calculated quantitatively, comparing the measured fluorescence intensity in the probe and the control. 3. The inference that S1 directly increases aggregation of α-Syn (when α-Syn level is sufficient) via induction of mitochondrial ROS is not correct because the action is indirect. A direct action of S1 could appear inside the neuronal cells where alpfa-Syn is located, but S1-subunit is a globular protein which can not penetrate through their cytoplasmic membranes. 4. The authors should give more information about the metformin (nature, chemical structure, molecular mass, etc) to facilitate the potential readers.
